# Current Management and Treatment Paradigms of Gastroesophageal Reflux Disease following Sleeve Gastrectomy

**DOI:** 10.3390/jcm13051246

**Published:** 2024-02-22

**Authors:** Muaaz Masood, Donald E. Low, Shanley B. Deal, Richard A. Kozarek

**Affiliations:** 1Division of Gastroenterology and Hepatology, Center for Digestive Health, Virginia Mason Franciscan Health, Seattle, WA 98101, USA; muaazm@gmail.com; 2Division of Thoracic Surgery, Center for Digestive Health, Virginia Mason Franciscan Health, Seattle, WA 98101, USA; donald.low@virginiamason.org; 3Division of General and Bariatric Surgery, Center for Weight Management, Virginia Mason Franciscan Health, Seattle, WA 98101, USA; shanley.deal@virginiamason.org; 4Center for Interventional Immunology, Benaroya Research Institute, Virginia Mason Franciscan Health, Seattle, WA 98101, USA

**Keywords:** gastroesophageal, reflux, GERD, esophagitis, morbid, obesity, bariatric, gastric, sleeve, gastrectomy

## Abstract

Obesity is associated with serious comorbidities and economic implications. Bariatric surgery, most commonly Roux-en-Y gastric bypass and sleeve gastrectomy, are effective options for weight loss and the improvement of obesity-related comorbidities. With the growing obesity epidemic, there has been a concomitant rise in bariatric surgeries, particularly in sleeve gastrectomy, which has been the most widely performed bariatric surgery since 2013. Gastroesophageal reflux disease (GERD) is highly prevalent in obese individuals, can significantly impact quality of life and may lead to serious complications. Obesity and GERD both improve with weight loss. However, as the incidence of sleeve gastrectomy rises, recent data have revealed a risk of exacerbation of pre-existing GERD or the development of de novo GERD following sleeve gastrectomy. We performed a detailed review of GERD post-sleeve gastrectomy, including its overall incidence, pathophysiology and current treatment paradigms.

## 1. Introduction

Obesity, defined as a body mass index (BMI) > 30 kg/m^2^, affects approximately four out of ten individuals in the United States and has serious health and economic implications [1]. The World Obesity Foundation has projected that over 4 billion people, or half of the world’s current population, will be overweight or obese by 2035 (World Obesity Federation, World Obesity Atlas, 2023). Obesity is associated with several chronic conditions including diabetes, hypertension, hyperlipidemia, obstructive sleep apnea, metabolic dysfunction-associated steatotic liver disease and malignancy [2,3,4]. Coinciding with the growing obesity epidemic, there has been an increase in the rate of bariatric surgeries over the last two decades [3]. Roux-en-Y gastric bypass (RYGB) and sleeve gastrectomy (SG) are the most commonly performed bariatric surgeries [3]. Sleeve gastrectomy surpassed RYGB in 2013 and continues to be the most widely performed bariatric surgery to date [3,5]. A shift towards minimally invasive bariatric surgery compared to open bariatric surgery has also been evident over the past 20 years [3].

Gastroesophageal reflux disease (GERD) is a chronic condition which is highly prevalent in obese individuals [6,7]. GERD manifests as heartburn and regurgitation although atypical or extraesophageal symptoms, in particular cough and chest pain, may be present in up to one-third of patients [8]. GERD has been documented in 62.4–73% of candidates for bariatric surgery [9,10]. Obesity and GERD both improve with weight loss [11,12]. However, as the incidence of minimally invasive SG increases, emerging data have revealed a risk of worsening GERD post-SG [5,13,14,15,16,17]. De novo GERD has also been reported after SG [13,14,18,19,20]. We performed a detailed review of GERD following SG, including its overall incidence, pathophysiology and current treatment paradigms.

## 2. Methods

We conducted a narrative review of the literature pertaining to GERD following sleeve gastrectomy using a PubMed and Google Scholar computerized search to identify articles with the title or keywords “GERD”, “gastroesophageal”, “reflux”, “obesity”, “bariatric surgery”, “laparoscopic”, “sleeve gastrectomy” and “gastric sleeve”. Systematic reviews, meta-analyses, randomized controlled trials and clinical trials were prioritized. Articles that were unrelated to the topic and duplicate articles were removed. Articles that were not available in English or full text were excluded. Articles regarding reflux in the context of pregnancy were excluded. This article is based on previously conducted studies and does not contain any new studies with human participants or animals performed by any of the authors. A total of 109 articles were included in the final review. The articles were independently reviewed by two authors. The authors provided representative, anonymized radiographic and endoscopic images to highlight the issues outlined in the manuscript.

## 3. Discussion

### 3.1. What Is the Evidence for Worsening GERD following SG?

Sleeve gastrectomy remains one of the most effective options for weight loss and the improvement of obesity-related comorbidities, namely diabetes, hypertension, hyperlipidemia and obstructive sleep apnea [2,3,4]. Sleeve gastrectomy also has a slightly lower postoperative complication rate when compared to RYGB [21]. As the incidence of SG rises, however, GERD following SG has become a notable concern. A multitude of studies have associated sleeve gastrectomy with the exacerbation of pre-existing GERD (Table 1). A retrospective review of 28 patients suggested that LSG may increase in the prevalence of GERD despite adequate weight loss [15]. Matar et al. retrospectively reviewed 517 patients and noted a higher prevalence of erosive esophagitis (EE) (37.9% vs. 17.6%, *p* = 0.0001), including severe EE, with SG vs. RYGB [16]. A decreased resting lower esophageal sphincter was also noted in the SG group [16]. A prospective, nonrandomized, open-label study of 30 patients noted GERD in two-thirds of patients following LSG [22]. A meta-analysis by Gu et al. exploring the correlation between bariatric surgery (LSG and RYGB) and GERD revealed that the risk of GERD was five times greater with LSG compared to LRYGB (pooled OR of 5.10, *p* < 0.001) [17]. LRYGB was demonstrated to be more effective for the treatment of GERD in obese patients in the study [17]. DuPree et al. revealed that only 15.9% of 4832 patients demonstrated GERD resolution after SG whereas the vast majority of patients continued to have GERD symptoms [18]. Preoperative GERD was associated with significantly increased postoperative complications, gastrointestinal adverse events and the need for revisional surgery in the study [18]. A retrospective study involving 176 patients revealed that 34.6% of patients had preoperative GERD complaints, 49% had GERD symptoms within 30 days following LSG and 47.2% had GERD symptoms for greater than one month after LSG [19]. A recent multicenter, randomized controlled study of 240 patients comparing outcomes in gastric sleeve vs. gastric bypass, termed the SLEEVEPASS trial, reported a significantly higher PPI intake (64% vs. 36%, *p* < 0.001), a higher GERD-HRQL total score (10.5 vs. 0, *p* < 0.001), more reflux symptoms and a higher prevalence of esophagitis (31% vs. 7%, *p* < 0.01) in the LSG group [23]. The SM-BOSS trial of 217 patients noted that 31.8% developed worsening GERD at 5 years based on the patients’ symptoms and medication usage. Moreover, a study by Raj et al. noted that up to 35% of patients may experience GERD following laparoscopic SG (LSG) [22].

### 3.2. Is De Novo GERD a Topic of Concern following Sleeve Gastrectomy?

In addition to worsening pre-existing GERD, new-onset or de novo GERD has been widely reported after SG, as detailed in Table 1. In a recent meta-analysis by Znamirowski et al., the pooled odds ratio (OR) for overall GERD occurrence post-LSG across eight studies and 599 patients was 3.61 (95% confidence interval: 1.92–6.79; *p* < 0.001) [31]. The combined percentage of patients with de novo GERD across five studies was 50.8% [31]. Across seven studies, the combined percentages of patients with LA Grade C and LA Grade D esophagitis were 4.3% and 3.3%, respectively [31]. However, significant heterogeneity was noted in the study [31]. In addition, there was variability in how GERD was documented, as studies used patients’ symptoms, PPI usage, endoscopic findings, pH testing, esophageal manometry or a combination of these parameters. A meta-analysis by Han et al. reported that the pooled risk ratio of de novo GERD for SG vs. RYGB was 0.33 (95% confidence interval: 0.15–0.68; *p* = 0.003) [28]. Additional meta-analyses by Yeung and Oor have reported rates of de novo GERD to be 20% and 23%, respectively [29,30]. Finally, the SM-BOSS trial noted that 31.6% of 217 patients developed de novo GERD at 5 years based on patients’ symptoms and medication usage [26].

Overall, these data highlight that there is a persistent concern for worsening and de novo GERD after SG. The risk of exacerbation of pre-existing GERD and the development of new-onset GERD should be considered when assessing patients for SG.

### 3.3. What Are the Underlying Mechanisms for De Novo or Increased GERD following a Sleeve Gastrectomy?

There are factors which may contribute to alleviating GERD after SG including weight loss leading to decreased abdominal pressure, reduced gastric volume and possibly accelerated gastric emptying [32,33]. In contrast, several pathophysiological mechanisms for de novo or increased GERD following SG have been described. Anatomic disruption of the antireflux barrier, including the esophagogastric junction, gastroesophageal flap valve, the angle of His, the gastric sling fibers and the fundus, may result in an increasing propensity for the reflux of gastric contents. Others discuss that caution must be made when positioning the staple line during SG in a manner that will preserve the antireflux barrier [34]. In contrast to RYGB, SG may be more likely to disrupt the angle of His and gastroesophageal junction attachments. There may be a functional impairment at the gastroesophageal junction such as with a hiatal hernia (Figure 1 and Figure 2). In one study, 37% of 181 morbidly obese patients were noted to have a hiatal hernia during preoperative work-up for bariatric surgery [35]. Repair of the hiatal hernia, if present, at the time of SG is recommended to reduce the risk of postoperative GERD although hiatal hernia repair may pose a challenge in patients with large hernias, especially those associated with esophagitis or Barrett’s esophagus [35,36].

Conditions in which there is an increase in intragastric pressure may lead to reflux. Greenan et al. explored gastric pressurization using high-resolution manometry and reflux using pH impedance in patients post-SG and post-RYGB as well as in symptomatic and asymptomatic control groups. Patients in the SG group had higher acid exposure time (median 6.0% vs. 0.2%), reflux episode numbers (median 63.0 vs. 37.5) and baseline intragastric pressure (median 17.3 mmHg vs. 13.1 mmHg) compared to the RYGB group, *p* < 0.001 [37].

Complications of SG, including stenosis, angulation or kinking, are associated with increased intragastric pressure and GERD (Figure 1) [38]. Data regarding the optimal bougie size for weight loss and reduced reflux are mixed, but generally support a range of 36–42 French [39,40,41,42,43]. Overfilling of the gastric sleeve due to a large meal portion may result in reflux symptoms, and thus, patients should be counseled to gradually advance their diet from liquids to small portions of healthy, protein-rich foods. The gastric sleeve dimensions should be sufficient, as too narrow of a gastric sleeve may result in increased intragastric pressure and subsequent reflux. An abnormally narrow sleeve may also lead to regurgitation due to overfilling, even in patients with a competent lower esophageal sphincter and no hiatal hernia. Felinska et al. described the ideal shape of a gastric sleeve to be a trapezoid with a wide antrum and a narrow cardia to prevent sleeve stenosis or increased pressure within the stomach [34]. Furthermore, the preservation of the antrum has been suggested to increase gastric emptying and reduce subsequent GERD [44]. Esophagogastric dysmotility may be a contributing factor to post-SG GERD and a dynamic contrast study can be a useful test in establishing the diagnosis.

### 3.4. What Are the Current Management and Treatment Paradigms of GERD following SG?

#### 3.4.1. Lifestyle Modifications

A comprehensive, stepwise approach to the management and treatment of GERD following SG has recently been published by Masood et al. [45]. Lifestyle modifications are initially recommended following SG as part of a comprehensive bariatric surgery center’s pre- and postoperative education and long-term follow-up plan with patients. Dietary education should emphasize the gradual progression of the postoperative diet from liquids to eventually healthy, protein-rich meals. Small portion sizes, typically ½ to 1 cup or 4 to 6 ounces, are suggested. Overfilling the stomach with a large portion may exacerbate reflux and induce regurgitation or food stasis in the distal esophagus, which may also be interpreted as GERD-like symptoms or induce stasis esophagitis if overfilling occurs repeatedly (Figure 3). Counseling regarding exercise, nutritional supplementation, the treatment of constipation and the avoidance of underlying risk factors of GERD, i.e., alcohol, tobacco, caffeine, chocolate and postprandial supination, should be provided.

#### 3.4.2. Optimization of Medication Regimens

Patients are often prescribed a PPI or histamine-2 receptor antagonist (H2RA) to prevent ulcer formation and reflux. In patients with post-SG GERD, PPI and H2RA regimens should be reviewed and optimized accordingly. Of note, the novel potassium-competitive acid blocker, vonoprazan, has shown promising results in PPI-refractory GERD, although its use has not been assessed in the bariatric surgery population [46,47,48]. 

#### 3.4.3. Diagnostic Work-Up

The authors recommend obtaining an esophagram, EGD, manometry and pH testing in selected patients presenting with GERD following SG for a further anatomic and physiologic evaluation of symptoms. Post-SG complications should be promptly identified and corrected. Stenosis, angulation and kinking of the gastric sleeve are associated with increased intragastric pressure and reflux (Figure 3) [38]. Ulcers and gastric sleeve leaks should be addressed appropriately. 

#### 3.4.4. Treatment of Underlying Postoperative Complications: Gastric Sleeve Stenosis

Endoscopic balloon dilation (EBD) has been shown to be a safe and efficacious modality for gastric sleeve stenosis (GSS) (Figure 4) [49,50]. A meta-analysis by Chang et al. of 18 studies involving a total of 426 patients revealed an overall success rate of 76% and an average of 1.8 dilations per patient [51]. Proximal GSS had a higher efficacy rate of 90% compared to distal GSS which had an efficacy rate of 70% [51]. Studies in the meta-analysis utilized through-the-scope, controlled radial expansion balloons, pneumatic balloons used for achalasia or a combination of the two [51]. The optimal balloon type, balloon size and the number of dilations required remains unclear [52]. Fully covered self-expanding metal stents (FCSEMS) have been reported to be effective in 70% of cases in which EBD failed though stent migration remains a concern [51,53].

#### 3.4.5. Treatment of Underlying Postoperative Complications: Gastric Sleeve Leak

It is noteworthy to mention gastric sleeve leak (GSL), a serious, life-threatening complication of SG, which involves the leak of gastrointestinal contents from the staple line into the abdominal cavity. GSL may occur at any point along the staple line from the gastroesophageal junction to the antrum, although they most commonly occur at the angle of His. GSL confers a high risk of morbidity and mortality. The leak rate has been reported to be between 1% and 5% for patients who undergo SG as a primary procedure. The leak rate is >10% for patients who underwent SG as a revisional procedure, after failed laparoscopic adjustable gastric banding or vertical banded gastroplasty, likely due to increased dissection, prior staple lines, vascular integrity and the overall quality of tissue adhesions which may be encountered during a reoperation [54,55,56,57]. 

Csendes and Burgos proposed a classification of GSL according to the timing, location and severity of the leak [55,58]. Type I (subclinical) leaks are well-defined without septic complications or dissemination into the abdominal or pleural cavity. Type II leaks (clinical) are outlined by septic complications with generalized dissemination into the abdominal or pleural cavity. Leaks may be classified based on timing as early, intermediate and late if they occur 1–3 days, 4–7 days and ≥8 days following surgery, respectively. Leaks may also be classified using a combination of clinical and radiologic data as follows: Type A are microperforations without clinical or radiologic evidence of leak; Type B are leaks noted on radiographic studies without a clinical finding; and Type C are leaks with both radiologic and clinical evidence [59]. GSL may manifest as sepsis, abdominal pain, chest/shoulder pain and dizziness or it can be asymptomatic. Radiographic imaging, including upper gastrointestinal series and CT with contrast, are typically helpful in establishing the diagnosis. 

The management of GSL may include initial measures (i.e., broad-spectrum IV antibiotics, IV fluids, NPO status, distal enteral tube feeding), surgical consultation, early oversewing and open or laparoscopic drainage [57]. In recent years, several endoscopic techniques have been utilized in the management of GSL, including covered SEMS (partially covered or fully covered), internal drainage using transgastric double pigtail stents, over-the-scope clips, endoscopic suturing and the use of sealants, i.e., fibrin glue or cyanoacrylate [60]. 

A novel endoscopic modality known as endoscopic vacuum therapy (EVT) involves the placement of a wound vac sponge near the leak site. The sponge is attached to a nasogastric tube and continuous suction is applied using a wound vac device, which aids in the removal of fluid, pus and necrotic debris from the site [61]. The sponge is typically replaced endoscopically every 3 to 7 days. Additionally, patients often require a percutaneous endoscopic jejunostomy tube for enteral nutrition while the sponge is in place. Several studies report that EVT is safe and effective [62,63,64]. Although its feasibility as an initial modality for the treatment of sleeve leak is limited, given that it typically requires multiple endoscopic procedures, there may be a role for EVT in refractory cases of sleeve leak [60]. Ultimately, some patients who have failed prior therapies for sleeve leaks undergo conversion to RYGB or total gastrectomy [57].

#### 3.4.6. Discussion of Risks Related to GERD with Potential Candidates for Sleeve Gastrectomy

Due to the association of SG with an increased risk of GERD, it is paramount for clinicians to discuss the risk of exacerbation of pre-existing GERD and the development of new-onset GERD during the shared decision-making process with potential candidates for SG (Figure 5). Data regarding the long-term risk of PPI use and the risk of cardiovascular adverse events should be considered as they relate to the treatment of GERD post-SG.

An international expert panel on sleeve gastrectomy reported that 52.5% of general surgeons and 23.3% of bariatric surgeons considered GERD as a contraindication to SG [65]. According to a multi-society consensus statement, SG should not be performed as an antireflux procedure [66]. Data suggest that patients with a BMI > 35 and medically refractory GERD should be considered for RYGB rather than fundoplication due to the following: an increased risk of hiatal hernia and GERD recurrence with fundoplication in patients with BMI > 35, as well as an additional weight loss and metabolic benefit conferred by RYGB to address a myriad of comorbidities [66,67,68,69,70]. 

The authors recommend a thorough preoperative evaluation to include a detailed history of reflux-related symptoms in addition to pertinent objective testing, including esophagram, EGD, manometry and pH testing, in selected patients with GERD for a further anatomic and physiologic evaluation of symptoms. If there is evidence of a hiatal hernia or an incompetent esophagogastric junction, the authors would caution against performing a sleeve gastrectomy alone without a concomitant antireflux procedure. If there is evidence of Barrett’s epithelium preoperatively, RYGB should be considered. Further studies are warranted to define the optimal selection criteria for sleeve gastrectomy. 

#### 3.4.7. Conversion to Roux-en-Y Gastric Bypass

Conversion from SG to RYGB is ultimately necessary for some patients with refractory GERD post-SG (Figure 6). There are robust data that conversion to RYGB is very effective for the treatment of GERD, as detailed in Table 2, in addition to its beneficial effects on weight loss and other comorbidities (i.e., diabetes, hypertension, dyslipidemia). In a recent retrospective study by MacVicar et al., 4412 patients required revisional surgery due to GERD, which comprises 24% of all conversion procedures [71]. SG was the original surgery in 80.1% of cases and Roux-en-Y was the revisional surgery in 84.4% of cases [71]. However, conversion from SG to RYGB is not without additional risk. In a recent study with matched cohorts of 5912 patients, an increase in re-admissions, intervention, conversion to open surgery and operative time was documented in the group which underwent conversion from SG to RYGB compared to primary RYGB. There were no significant differences in mortality or complications noted between the two groups. In a study by Dang et al., conversion from SG to RYGB compared to primary RYGB was associated with an increased rate of serious complications (7.2% vs. 5%, *p* < 0.001), including anastomotic leaks, bleeding and reoperation, although no significant difference in mortality between the two groups was demonstrated in the study [72].

Many studies reveal significant improvement in GERD symptoms and the use of acid suppression medications following conversion to RYGB. Studies by Langer et al. and Gautier et al. both reported that all patients were able to discontinue acid suppressive medications [73,74]. Parmar et al. revealed that 80% of patients discontinued acid suppressive medications whereas Hendricks et al. documented that 75% of patients had a complete resolution of GERD with conversion to RYGB [75,76]. A study by Strauss et al. concluded that 80.2% of patients who underwent conversion to RYGB had improvement in GERD symptoms [77]. 

The average rate of conversion from SG to RYGB due to GERD is approximately 5–10% [26,30,78]. In the SLEEVEPASS trial, the rate of conversion to RYGB was noted to be 6% [78]. In the SM-BOSS trial, the reported rate of conversion to RYGB was 9% [26]. In a meta-analysis of 46 studies by Yeung et al., 4% of patients underwent conversion to RYGB [30]. In a review of 73 cases, Langer et al. reported a conversion rate of 11% [73]. Some studies report a higher rate of conversion, although these studies had limitations [75,77,79]. 

In addition, Han et al. revealed that RYGB may be superior to SG with regard to GERD improvement, with a pooled risk ratio of 1.48 (95% confidence interval: 1.07–2.04; *p* = 0.02) [28].

**Table 2 jcm-13-01246-t002:** Comparison of studies exploring the effect of conversion from sleeve gastrectomy to Roux-en-Y gastric bypass as it relates to gastroesophageal reflux disease. Abbreviations: ASM = acid suppression medications, BE = Barrett’s esophagus, EE = erosive esophagitis, GERD = gastroesophageal reflux disease, NR = not reported, OR = odds ratio, PPI = proton-pump inhibitor, RYGB = Roux-en-Y gastric bypass, SG = sleeve gastrectomy.

First Author	Year Published	Article Type	Number of Cases or Studies	Conversion Rate to RYGB for GERD	Effect on GERD and Use of Acid Suppression Medications
Yeung [30]	2020	Systematic review and meta-analysis	46 studies	4%	NR
Salminen [78]	2018	Randomized controlled trial	240 cases	6%	NR
Peterli [26]	2018	Randomized controlled trial	217 cases	9%	NR
Langer [73]	2010	Retrospective review	73 cases	11%	100% with severe reflux discontinued ASM
Parmar [75]	2017	Prospective study	22 cases	45.5%	100% reported improvement in GERD symptoms 80% were able to discontinue ASM
Abdemur [80]	2016	Retrospective review	1118 cases	0.8%	66% had complete resolution of GERD symptoms
Hendricks [76]	2016	Retrospective review, comparative study	919 cases	10.5%	75% of patients had complete resolution of GERD symptoms 25% of patients had partial resolution of GERD symptoms
Gautier [74]	2013	Retrospective review	481 cases	1.24%	100% of patients discontinued ASM No recurrence of GERD was noted
Strauss [77]	2023	Retrospective review	97 cases	72.2%	80.2% of patients had improvement in GERD symptoms 19.4% of patients were able to discontinue ASM
Felsenreich [79]	2022	Retrospective review	79 cases	34.2%	29.9% of patients reported GERD symptoms following conversion
Peng [81]	2020	Systematic review and meta-analysis	40 studies	NR	57.1–100% had remission or improvement in GERD symptoms

#### 3.4.8. Sleeve Gastrectomy with Fundoplication or Hiatal Hernia Repair

Due to the prevalence of hiatal hernias in the bariatric population, several studies have explored GERD outcomes when SG is combined with either fundoplication or hiatal hernia repair (Figure 7 and Figure 8) [82,83,84,85,86,87,88,89,90,91,92,93,94]. The authors acknowledge that the published studies on SG combined with either fundoplication or hiatal hernia repair are investigational, with a limited sample size and variable follow-up. We summarize the published data regarding SG combined with either fundoplication or hiatal hernia repair to highlight experimental approaches to GERD post-SG beyond conversion to RYGB with hiatal hernia repair.

A randomized clinical trial comparing SG with SG and Rosetti fundoplication (RF) by Olmi et al. concluded that there was a significant reduction in PPI use and esophagitis in the SG with RF group compared to the SG group [88]. Wrap perforation occurred in 4.3% of patients and reportedly improved with the surgeon’s learning curve [88]. In a study of 220 patients who underwent SG with modified RF and a follow-up period of 24 months, 98.5% of cases did not report GERD or use PPIs and 97% of cases had endoscopic improvement in esophagitis [89]. A prospective study of 25 patients who underwent laparoscopic Nissen-SG (N-Sleeve) concluded that 76% of patients remained asymptomatic without requiring PPI at 3 months [95]. Twelve percent of patients reported reflux at 6 months and 12 months postoperatively [95]. In a study of 32 patients with GERD/esophagitis who underwent SG with anterior fundoplication (D-Sleeve), a comparison of high-resolution impedance manometry and multichannel intraluminal impedance before and after the D-sleeve surgery revealed improved lower esophageal sphincter function and notable control of esophageal acid exposure and reflux events at 14 months [92].

A systematic review and meta-analysis by Castagneto-Gissey et al. compared the efficacy of SG with fundoplication vs. SG with hiatal hernia repair as it relates to GERD remission across 15 studies, with a total of 1164 patients [82]. The mean follow-up was 37.3 ± 28.1 months after SG with hiatal hernia repair and 17.4 ± 9.3 months after SG with fundoplication [82]. Most studies defined GERD based on patients’ symptoms, with severity evaluated using scales or scores, and the use of antacids or PPIs [82]. Five studies in the meta-analysis utilized an objective assessment including endoscopy, manometry or upper gastrointestinal contrast studies to diagnose GERD [82]. Both SG with hiatal hernia repair and SG with fundoplication were effective in GERD resolution and weight loss outcomes [82]. However, SG with RF resulted in significantly greater GERD remission compared to SG with hiatal hernia repair although a higher complication rate was noted in the SG with RF group [82]. SG with fundoplication has been described to be relatively safe with a postoperative complication rate of 9.4% in a review by Carandina et al. [83]. While there are preliminary data regarding SG with fundoplication, the authors conclude that additional, high-quality studies are warranted to determine the long-term safety and efficacy of SG with fundoplication.

#### 3.4.9. The Use of Magnetic Sphincter Augmentation Devices

Magnetic sphincter augmentation (MSA) devices, such as the LINX© system, have also been used as an experimental approach to mitigate post-SG GERD. MSA devices utilize titanium beads with a magnetic core connected with titanium wires to form a ring shape, which is laparoscopically placed around the lower esophageal sphincter as a reinforcement (Figure 9 and Figure 10). Some studies have revealed favorable results [96,97,98,99,100]. A meta-analysis of three studies with a total of 33 patients revealed a pooled mean difference between preoperative and postoperative GERD-Heath-Related Quality of Life Questionnaire (GERD-HRQL) scores to be 17.5 following MSA [101]. However, data are limited. Further studies are warranted to establish the safety and efficacy of MSA devices following SG.

#### 3.4.10. Endoscopic Sleeve Gastroplasty

Of note, a recent technique known as endoscopic sleeve gastroplasty (ESG), or endoluminal vertical gastroplasty (EVG), has been reported to be a safe and effective primary weight loss procedure or revisional procedure for weight recidivism in the setting of SG and has also shown promising results with regard to GERD improvement [102,103,104,105]. According to a case-matched study by Fayad et al., new-onset GERD was significantly lower in the ESG group compared with the LSG group (1.9% vs. 14.5%, *p* < 0.05), which may be due to sparing of both the fundus of the stomach and attachments to the gastroesophageal junction [106]. In a multicenter study of 18 patients, the Transoral Gastric Volume Reduction as Intervention for Weight Management (TRIM) trial, which evaluated outcomes in transoral gastric volume reduction surgery using an endoscopic suturing system, reported a reduction in patients who reported reflux from 8 of 14 patients prior to the procedure to 5 out of 14 at 1-year follow-up [107]. Further studies are needed to determine the effect of ESG on GERD.

#### 3.4.11. Surveillance for Barrett’s Esophagus Post-Sleeve Gastrectomy

It is important to note that emerging data have revealed an association of Barrett’s esophagus (BE) following SG. According to two recent meta-analyses by Yeung and Qumseya, the prevalence of BE following SG is approximately 8% and 11.6%, respectively [30,108]. While the current data regarding BE following SG have limitations, they suggest that endoscopic surveillance is important following SG. The American Society for Metabolic and Bariatric Surgery (ASMBS) has recommended to offer screening for BE in SG patients three or more years post-SG, regardless of the presence of GERD symptoms, in addition to the standard screening indications for GERD and Barrett’s esophagus [109].

## 4. Conclusions

With the growing obesity epidemic and concomitant rise in bariatric surgery, particularly sleeve gastrectomy, there are robust data that sleeve gastrectomy is associated with an exacerbation of pre-existing GERD and the development of de novo GERD due to several pathophysiological mechanisms. Current management and treatment paradigms should include the following: lifestyle modifications, optimization of PPI and H2RA regimens, diagnostic work-up (i.e., esophagram, EGD, manometry and pH testing) as indicated in select patients and prompt endoscopic or surgical treatment of postoperative complications (i.e., gastric sleeve stenosis, angulation, kinking, leaks). Conversion to RYGB currently has the most robust data for its safety and efficacy for the treatment of medically refractory GERD post-SG. Discussion and shared decision making regarding the risk of worsening or new-onset GERD with potential candidates for sleeve gastrectomy are paramount. Data suggest that patients with a BMI > 35 and medically refractory GERD should be considered for RYGB rather than fundoplication. Due to the reports of Barrett’s esophagus following sleeve gastrectomy, the ASMBS recommends that endoscopic surveillance should be offered to patients three or more years following sleeve gastrectomy, regardless of the presence of GERD symptoms, in addition to standard screening indications for GERD and Barrett’s esophagus. Further studies are warranted to determine the role of sleeve gastrectomy combined with fundoplication or hiatal hernia repair, endoscopic sleeve gastroplasty and the use of magnetic sphincter augmentation devices in relation to GERD.

## Figures and Tables

**Figure 1 jcm-13-01246-f001:**
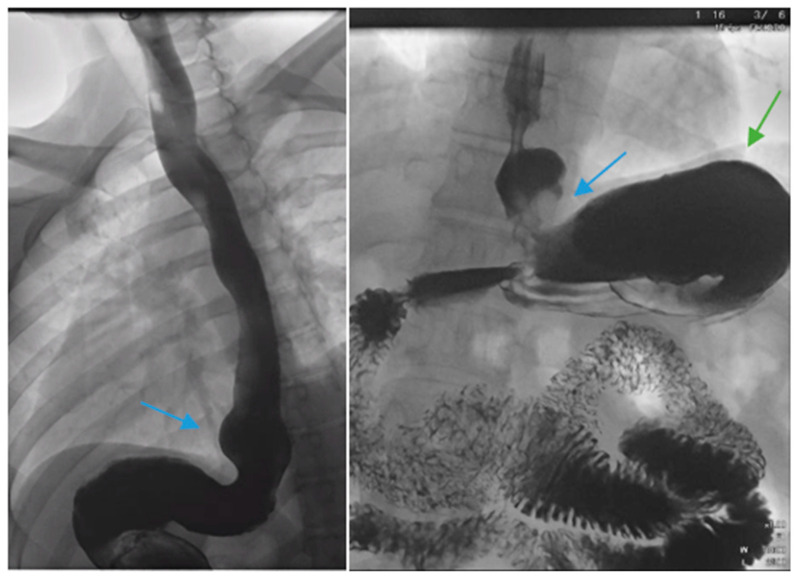
Upper gastrointestinal series in a patient post-sleeve gastrectomy with early satiety and reflux reveals the following: a small, hiatal hernia and gastroesophageal reflux (blue arrow, **left**) in addition to abnormal angulation (blue arrow, **right**) and dilation of the gastric sleeve (green arrow, **right**). The patient underwent conversion to Roux-en-Y gastric bypass with marked clinical improvement.

**Figure 2 jcm-13-01246-f002:**
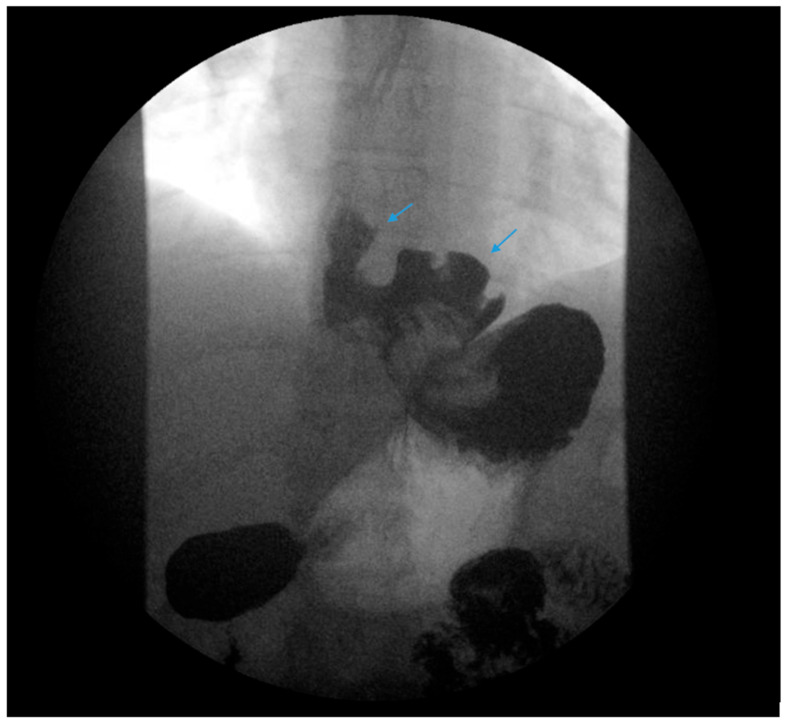
An upper gastrointestinal series reveals a moderate, sliding type-one hiatal hernia (bottom arrow) in addition to gastroesophageal reflux disease (top arrow) in a patient with morbid obesity.

**Figure 3 jcm-13-01246-f003:**
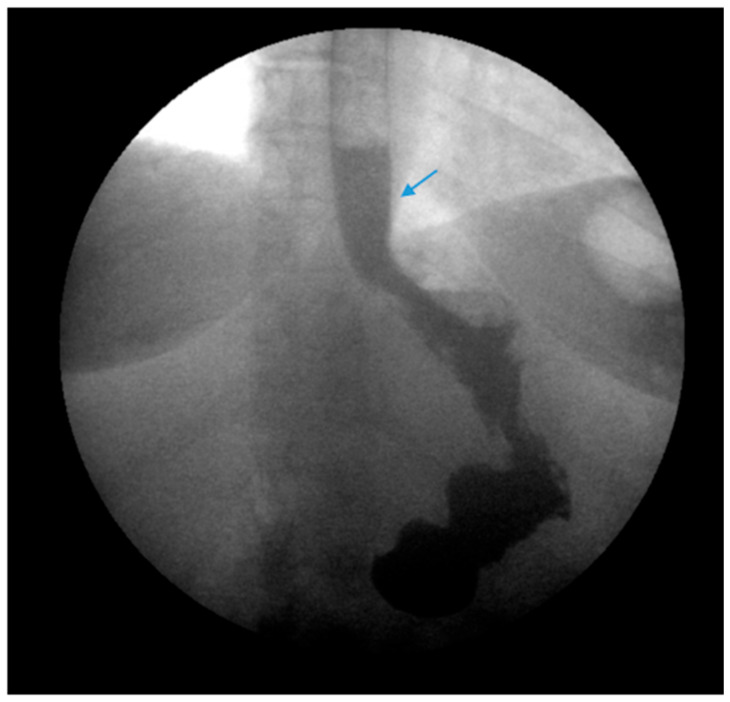
An upper gastrointestinal series reveals contrast filling of sleeve with reflux of a column of contrast into the distal esophagus (blue arrow) in a patient following sleeve gastrectomy.

**Figure 4 jcm-13-01246-f004:**
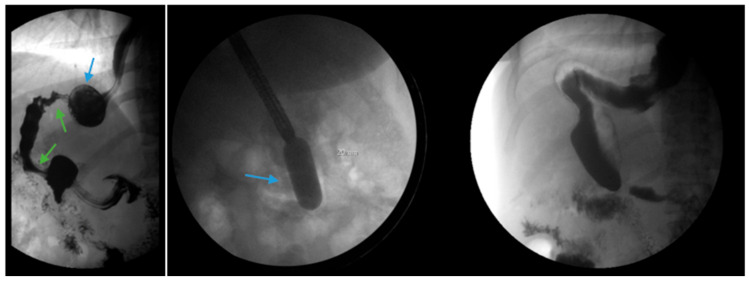
(**Left**) An upper gastrointestinal series in a patient post-sleeve gastrectomy with recurrent dysphagia reveals dilation of the gastric sleeve at the cardia (blue arrow) with narrowing and delayed passage of contrast through the middle and distal portions (green arrows) of the gastric sleeve. (**Middle**) The patient underwent serial endoscopic through-the-scope balloon dilation to 20 mm (blue arrow) with improvement in sleeve stenosis (**Right**) and dysphagia.

**Figure 5 jcm-13-01246-f005:**
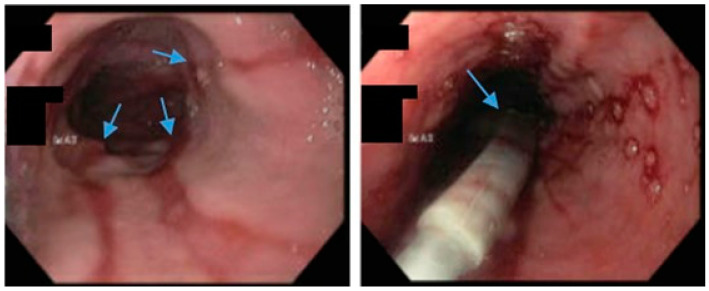
Upper endoscopy demonstrates Los Angeles Grade C esophagitis in a patient following sleeve gastrectomy (arrows, **left**). A Bravo™ capsule was placed endoscopically for pH monitoring (Medtronic, Minneapolis, Minnesota, USA) (arrow, **right**).

**Figure 6 jcm-13-01246-f006:**
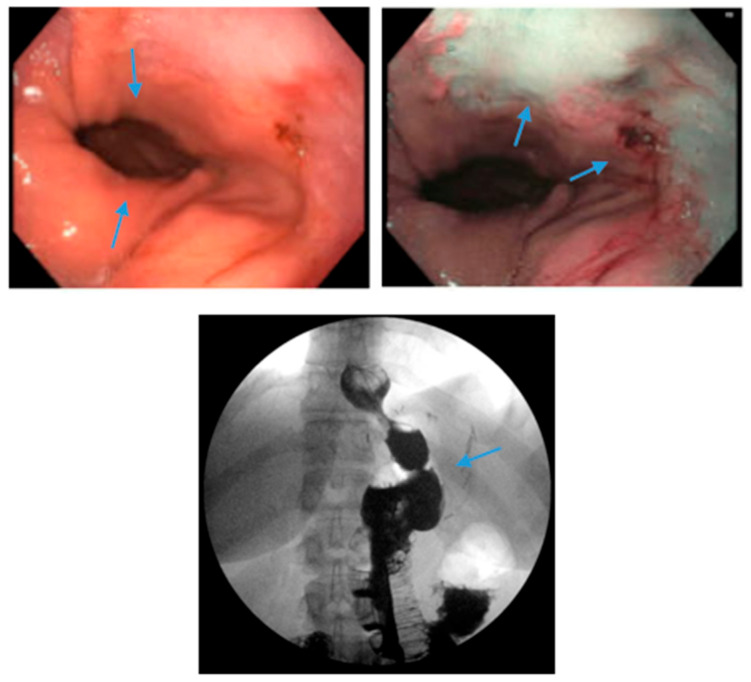
Endoscopic images reveal a 3 cm hiatal hernia (arrows, **top left**) and Los Angeles Grade B esophagitis (arrows, **top right**) in the lower third of the esophagus with narrow-band imaging in a patient post-sleeve gastrectomy with symptoms of gastroesophageal reflux disease. The patient eventually required Roux-en-Y gastric bypass (arrow, **bottom**) and concomitant hiatal hernia repair, as demonstrated on upper gastrointestinal series.

**Figure 7 jcm-13-01246-f007:**
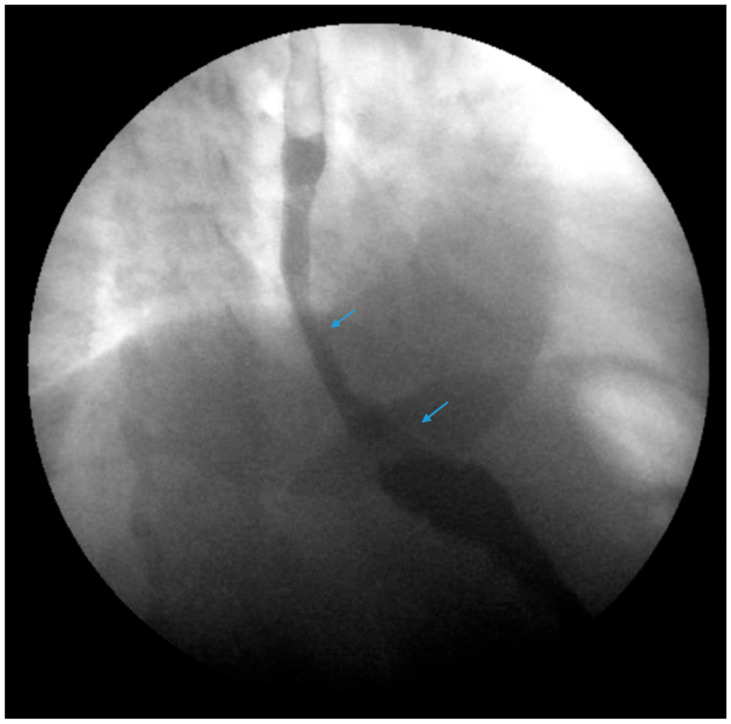
An upper gastrointestinal series reveals a small, sliding hiatal hernia (bottom arrow) in addition to gastroesophageal reflux disease (top arrow) in a patient following sleeve gastrectomy.

**Figure 8 jcm-13-01246-f008:**
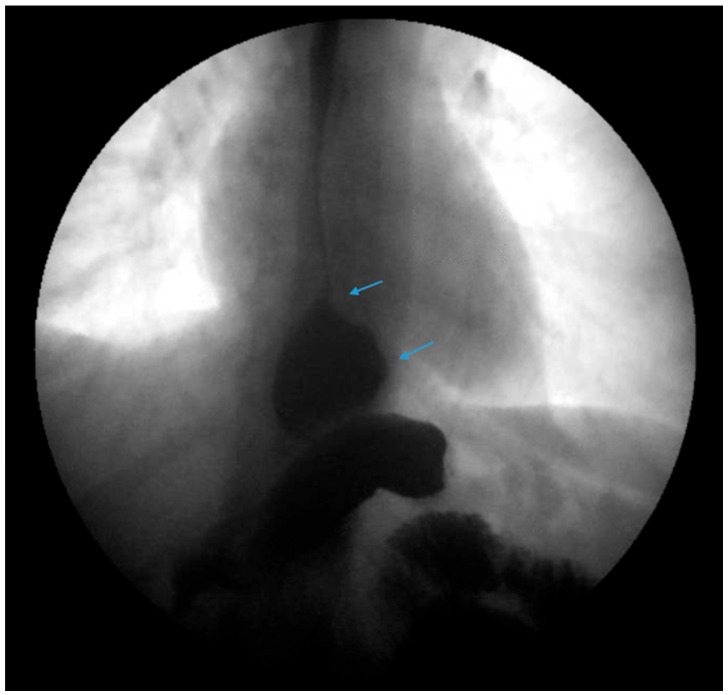
An upper gastrointestinal series reveals a small, sliding hiatal hernia (bottom arrow) with spontaneous gastroesophageal reflux (top arrow) in a patient following sleeve gastrectomy.

**Figure 9 jcm-13-01246-f009:**
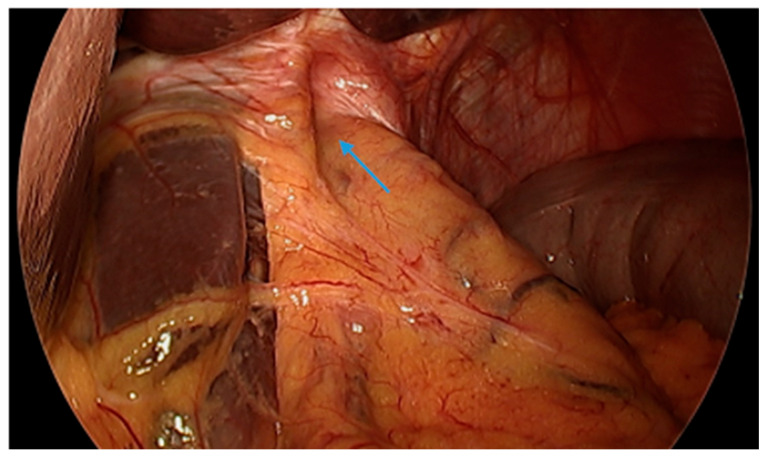
Laparoscopic image reveals a patient post-sleeve gastrectomy with a defect in the diaphragmatic hiatus (arrow) and subsequent gastroesophageal reflux disease. Image courtesy of Dr. Brian Louie, Chief of Thoracic Surgery at Swedish Medical Center in Seattle, Washington, USA.

**Figure 10 jcm-13-01246-f010:**
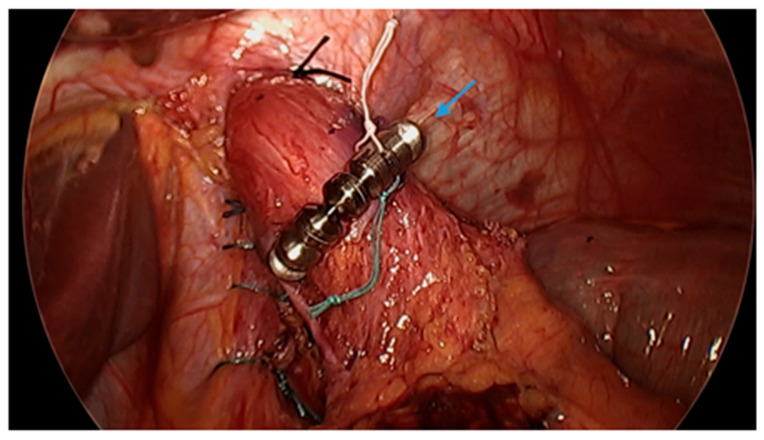
Laparoscopic image demonstrates hiatal hernia repair and successful placement of a magnetic sphincter augmentation device (arrow) around the esophagus in a patient post-sleeve gastrectomy with gastroesophageal reflux disease. Image courtesy of Dr. Brian Louie, Chief of Thoracic Surgery at Swedish Medical Center in Seattle, Washington, USA.

**Table 1 jcm-13-01246-t001:** Comparison of de novo and pre-existing gastroesophageal reflux disease/erosive esophagitis and acid suppression medication use in the setting of sleeve gastrectomy. Abbreviations: BE = Barrett’s esophagus, EE = erosive esophagitis, GERD = gastroesophageal reflux disease, NR = not reported, OR = odds ratio, PPI = proton-pump inhibitor, RYGB = Roux-en-Y gastric bypass, SG = sleeve gastrectomy, vs. = versus, → = indicates the comparison of a parameter before and after intervention, namely the specified bariatric surgery.

First Author	Year Published	Article Type	Number of Cases or Studies	Rate of De Novo or New-Onset GERD	Comparison of GERD or Erosive Esophagitis in the Setting of SG	Comparison of Acid Suppression Medication Use in the Setting of SG	Additional Comments
Tai [14]	2013	Retrospective review	66 cases	44.8%	GERD: 12.1% (pre-SG) → 47% (post-SG)	NR	Increase in hiatal hernias (6.1% pre-SG → 27.3% post-SG)
Gu [17]	2019	Systematic review and meta-analysis	23 studies	NR	GERD post-SG vs. post-RYGB, OR = 5.10, *p* < 0.001	NR	
Howard [15]	2011	Retrospective review	28 cases	18%	NR	NR	
Matar [16]	2020	Retrospective review	517 cases	NR	Prevalence of EE higher post-SG (37.9%) vs. post-RYGB (17.6%), *p* = 0.0001	NR	
Carter [19]	2011	Retrospective review	176 cases		34.6% (pre-SG) 49% (post-SG at 30 days) 47.2% (post-SG > 30 days)	22% (pre-SG) → 33.8% (post-SG), *p* = 0.0428	
DuPree [18]	2014	Retrospective review	4832 cases	8.6%	GERD pre-SG: 44.5% 84.1% of post-SG cases with continued GERD	NR	
Shepppard [13]	2015	Retrospective review	387 cases	NR	NR	28% pre-SG PPI use Only 2% of cases were able to discontinue PPI post-SG	
Joudeikis [24]	2017	Systematic review and meta-analysis	20 studies	NR	GERD improved in 30.6% of patients	NR	
Mion [25]	2016	Retrospective review	53 cases	NR	Frequent, impedance reflux episodes (52%) post-SG associated with GERD symptoms and esophageal dysmotility	NR	
Peterli [26]	2018	Randomized controlled trial	217 cases	31.6%	31.8% had worsening of symptoms or increase in therapy at 5 years post-SG		
Genco [27]	2017	Prospective review	110 cases	NR	33.6% (pre-SG) → 68.1% (post-SG), *p* < 0.001	PPI use: 19.1% (pre-SG) → 57.2% (post-SG), *p* < 0.001	Non-dysplastic BE was newly diagnosed in 17.2% of cases
Han [28]	2020	Systematic review and meta-analysis	20 studies	Pooled RR of de novo GERD with SG vs. RYGB was 0.33 (95% CI: 0.15 to 0.68, *p* = 0.003)	Pooled RR of GERD symptoms with SG vs. RYGB was 0.16 (95% CI: 0.06 to 0.44; *p* = 0.0004)	NR	
Oor [29]	2016	Systematic review and meta-analysis	33 studies	20%	New-onset esophagitis: 6.3–63.3%	NR	High heterogeneity among studies
Salminen [23]	2022	Randomized clinical trial	228 cases	NR	49% of cases reported worsening GERD post-SG at 10 years 31% had esophagitis at 10 years post-SG	PPI use: 64% at 10 years post-SG	De novo BE: 4%
Yeung [30]	2020	Systematic review and meta-analysis	46 studies	23%	23% increase in GERD post-SG	NR	Long-term prevalence of esophagitis was 28% and of BE was 8%
Znamirowski [31]	2023	Systematic review and meta-analysis	9 studies	50.8%	GERD: 26% (pre-SG) → 61.6% (post-SG)	NR	Significant heterogeneity among studies was reported BE was reported in 7.3% of cases across all studies

## Data Availability

The data availability statement is not applicable for this study. This manuscript has not been previously published and is not currently under consideration elsewhere for publication.

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
