# Peer review of "Current Management and Treatment Paradigms of Gastroesophageal Reflux Disease following Sleeve Gastrectomy"

_jcm, 2024, doi:10.3390/jcm13051246_

Round 1

Reviewer 1 Report

Comments and Suggestions for Authors

Dear authors,

I have carefully read your study and would like to congratulate you on its content. It is very well outlined and brings to the reader nearly all aspects involved with post-bariatric surgery gastroesophageal reflux following the sleeve gastrectomy (SG) technique. A well-done and well-illustrated narrative review, updating the possible ways to correct this frequently occurring syndrome. The text goes further, as beyond analyzing the reflux itself, it draws attention to possible technical or anatomical deviations that may have been caused by the surgery as contributors to the problem.

In my view, the work lacks a critical analysis of how we can better select patients for SG. Although it is not the focus of the study, the reader is left with this question unanswered. A BMI above 35 is a parameter surgeons adopt as an indication for bypass surgery. However, as we know, a BMI below 35 does not give us the right to perform the sleeve without running the risks of persistent previous reflux or new onset.

The consensus among surgeons is that many studies still lack evidence for a better selection of patients for this or that surgery, but certainly, clinical assessment, endoscopy, X-rays, manometric study, and pH impedance can aid in this decision.

Obesity is an epidemic, especially in the Western world and the Americas. The social cost of those untreated is very high! The cost of surgical treatment is also high. Reoperation for control of post-SG problems has higher costs and risks. Based on a reality recently published (O'Laughlin et al.), about 6883 patients undergoing Sleeve surgery will have to be reoperated in 2020-2021 in the USA, most of them probably due to the presence of gastroesophageal reflux that is untreatable clinically. We need to improve these numbers.

Along with the discussion and conclusions, there would be room to complement this excellent work with these considerations.

Warm regards

O'Laughlin M, Cornejo J, Zevallos A, Coker A, Schweitzer M, Adrales G, Li C, Sebastian R. Laparoscopic sleeve gastrectomy to Roux-en-Y gastric bypass conversion versus primary Roux-en-Y gastric bypass: a propensity score matching analysis. Surg Endosc. 2023 Oct;37(10):7947-7954. doi: 10.1007/s00464-023-10261-0. Epub 2023 Jul 11. PMID: 37433912.

Author Response

Thank you for your review and comments pertaining to our manuscript. The optimal selection criteria for a suitable candidate for sleeve gastrectomy is a question of important clinical relevance and remains a controversial topic among bariatric surgeons. The authors recommend a thorough preoperative evaluation to include a detailed history of reflux-related symptoms in addition to pertinent objective testing, i.e. esophagram, EGD, manometry and pH testing, in selected patients with GERD for further anatomic and physiologic evaluation of symptoms. If there is evidence of a hiatal hernia or an incompetent gastroesophageal junction, the authors would caution against performing a sleeve gastrectomy alone without a concomitant antireflux procedure. If there is evidence of Barrett’s epithelium preoperatively, RYGB should be considered. Further studies are warranted to define the optimal selection criteria for sleeve gastrectomy. The authors feel that additional elaboration on this topic would be outside the scope of our manuscript which speicifically serves to review the evidence for worsening and de novo GERD following sleeve gastrectomy.  The following sentences have been added to the manuscript to elaborate on this point.

“The authors recommend a thorough preoperative evaluation to include a detailed history of reflux-related symptoms in addition to pertinent objective testing, i.e. esophagram, EGD, manometry and pH testing, in selected patients with GERD for further anatomic and physiologic evaluation of symptoms. If there is evidence of a hiatal hernia or an incompetent gastroesophageal junction, the authors would caution against performing a sleeve gastrectomy alone without a concomitant antireflux procedure. If there is evidence of Barrett’s epithelium preoperatively, RYGB should be considered. Further studies are warranted to define the optimal selection criteria for sleeve gastrectomy.”

Additionally, the study by O’Laughlin et al. was incorporated in the “Conversion to RYGB” section as follows:

“However, conversion from SG to RYGB is not without additional risk. In a recent study of matched cohorts of 5912 patients, an increase in readmissions, intervention, conversion to open and operative time was documented in the group which underwent conversion from SG to RYGB compared to primary RYGB. There were no significant differences in mortality or complications noted between the two groups.

Please note, the following sentence, which was originally later in the section, was moved to after the above sentence as the study by Dang et al. also discussed outcomes with conversion from SG to RYGB:

“In a study by Dang et al., conversion from SG to RYGB compared to primary RYGB was associated with an increased rate of serious complications (7.2%% vs. 5%, p <0.001), including anastomotic leaks, bleeding and reoperation, although no significant difference in mortality between the two groups was demonstrated in the study [72].”

Reviewer 2 Report

Comments and Suggestions for Authors

This is an excellent meta-analysis, which discusses in detail the importance of reflux post-sleeve gastrectomy. I could find no errors.  The discussion is detailed and appropriate. It may have been good to have seen some positives about the sleeve, eg the slightly complication rate than RYGB, etc. However, overall an excellent paper with good discussion.

Author Response

Thank you for your review and positive feedback. Two sentences have been included in the manuscript which positively describe the efficacy of sleeve gastrectomy as it relates to weight loss, improvement in some of the weight-related comorbidities and its slightly lower postoperative complication rate when compared to RYGB as follows:

Sleeve gastrectomy remains one of the most effective options for weight loss and improvement of obesity-related comorbidities, namely diabetes, hypertension, hyperlipidemia and obstructive sleep apnea [2-4]. Sleeve gastrectomy also has a slightly lower postoperative complication rate when compared to RYGB [21].”

Additionally, please note the following sentence which was added in the Introduction section linking obesity to its comorbidities:

“Obesity is associated with several chronic conditions including diabetes, hypertension, hyperlipidemia, obstructive sleep apnea, metabolic dysfunction-associated steatotic liver disease and malignancy [2-4].”

Reviewer 3 Report

Comments and Suggestions for Authors

Thank you for involving me in the review of this excellent manuscript. Here are my comments:

  1. Section 3.1: Since studies with larger sample size or prospective design (ref 14, 19, 15) are quoted in this section, I think the smaller studies (ref 13, 14) and the associated text can be removed from the text and perhaps kept in the table for completeness of literature review. 
  2. Section 3.1, line 76- I dont think the abbreviation EE was introduced before being used in the text.
  3. section 3.2, OR 3.61 - please mention n and CI.
  4. section 3.2 - again a lot of this information is available in the tables and does not have to be duplicated in the text.
  5. Table 2: In Parmar and Strauss, the rates of conversion to RYGB from LSG are markedly different (45% and 72%) from the other studies (mostly <10%). Could the authors explain the same?
  6. Overall I felt that in a lot of places, the authors decided to expand the information in text, which is already discussed in tables. I think that can be made significantly concise to improve the readability of the manuscript.

Author Response

Thank you for your review and comments. Reference #13 and 14 have been removed from Section 3.1 text. If you wish, the following sentence associated with Reference #15 may also be removed from Section 3.1, but retained in Table 1 to shrink this section:

“A retrospective review of 28 patients suggested that LSG may increase in the prevalence of GERD despite adequate weight loss [15].”

The abbreviation “EE” has been defined as “erosive esophagitis” in its first occurence in Section 3.1

The number of patients in the study and the confidence interval for the Znamirowski et al. study has been incorporated in Section 3.2.

Only large meta-analyses and a pivotal clinical trial (SM-BOSS) are discussed in Section 3.2 text and thus, the authors would like to retain this information although it is summarized in Table 2.

Regarding the variance in the conversion rates to RYGB from SG, the studies by Parmar et al. and Strauss et al. were limited due to small sample size and retrospective design, respectively. The other referenced study with a higher conversion rate to RYGB from SG (Felsenreich et al.) also had a retrospective design. Thus, the following modifications were made:

“In a review of 73 cases, Langer et al. reported a conversion rate of 11% [73]. Some studies report a higher rate of conversion although these studies had limitations [75, 77, 79].”